# The Resection Rate of Synchronously Detected Liver and Lung Metastasis from Colorectal Cancer Is Low—A National Registry-Based Study

**DOI:** 10.3390/cancers15051434

**Published:** 2023-02-23

**Authors:** Jennie Engstrand, Helena Taflin, Jenny Lundmark Rystedt, Oskar Hemmingsson, Jozef Urdzik, Per Sandström, Bergthor Björnsson, Kristina Hasselgren

**Affiliations:** 1Division of Surgery, Department of Clinical Science, Intervention and Technology (CLINTEC), Karolinska Institutet, Karolinska University Hospital, 141 86 Stockholm, Sweden; 2Department of Surgery, Institute of Clinical Sciences, Sahlgrenska Academy at University of Gothenburg, Sahlgrenska University Hospital, 413 45 Göteborg, Sweden; 3Department of Surgery, Skåne University Hospital, Lund University, 221 00 Lund, Sweden; 4Department of Surgical and Perioperative Sciences, Umeå University, 901 87 Umeå, Sweden; 5Wallenberg Centre for Molecular Medicine, Umeå University, 901 87 Umeå, Sweden; 6Department of Surgical Sciences, Uppsala University, 752 36 Uppsala, Sweden; 7Department of Surgery in Linköping, Linköping University, 581 83 Linköping, Sweden; 8Department of Biomedical and Clinical Sciences, Linköping University, 581 83 Linköping, Sweden

**Keywords:** liver metastases, lung metastases, colorectal cancer, synchronous metastases, incidence, treatment

## Abstract

**Simple Summary:**

Real-life data on the occurrence and treatment of synchronously detected liver and lung metastases from colorectal cancer are lacking. Through the merging of several Swedish nationwide patient quality registries, we aimed to answer these questions. We found that synchronous liver and lung colorectal metastases are rare and that a minority undergo resection of both metastatic sites, but if they do, they have an excellent survival. It is likely that a larger proportion of patients could be offered treatment that leads to a prolonged overall survival. We also found differences in regional treatment approaches across Sweden, but the reasons for this are unknown, which warrants further studies.

**Abstract:**

Population-based data on the incidence and surgical treatment of patients with colorectal cancer (CRC) and synchronous liver and lung metastases are lacking as are real-life data on the frequency of metastasectomy for both sites and outcomes in this setting. This is a nationwide population-based study of all patients having liver and lung metastases diagnosed within 6 months of CRC between 2008 and 2016 in Sweden identified through the merging of data from the National Quality Registries on CRC, liver and thoracic surgery and the National Patient Registry. Among 60,734 patients diagnosed with CRC, 1923 (3.2%) had synchronous liver and lung metastases, of which 44 patients had complete metastasectomy. Surgery of liver and lung metastases yielded a 5-year OS of 74% (95% CI 57–85%) compared to 29% (95% CI 19–40%) if liver metastases were resected but not the lung metastases and 2.6% (95% CI 1.5–4%) if non-resected, *p* < 0.001. Complete resection rates ranged from 0.7% to 3.8% between the six healthcare regions of Sweden, *p* = 0.007. Synchronous liver and lung CRC metastases are rare, and a minority undergo the resection of both metastatic sites but with excellent survival. The reasons for differences in regional treatment approaches and the potential of increased resection rates should be studied further.

## 1. Introduction

Colorectal cancer (CRC) is the fourth most frequently diagnosed cancer and second leading cause of cancer-related death in Sweden [1]. About 15–25% of all CRC patients have distant metastases at the time of diagnosis of primary tumor in the colon or rectum, named synchronous metastases, where liver metastases are found in 17% and lung metastases are found in 5% [2,3]. Even though liver and lungs are the two most frequent sites of distant metastatic spread from CRC [4], the simultaneous diagnosis of both liver and lung metastases synchronously to CRC is not as frequent. A Dutch nationwide population-based study concluded that simultaneous liver and lung metastases were present in only 3.4% of all patients diagnosed with CRC between 2008 and 2011 [2].

The general assumption is that the complete metastasectomy of liver and lung metastases from CRC is oncologically beneficial [5,6]. The few studies on the surgical management of simultaneously diagnosed liver and lung metastases include both synchronously and metachronously diagnosed metastases and report a 5-year survival of 43–72% if all the intended metastasectomies are completed [7,8,9,10]. For comparison, the overall median survival for the entire group of patients with synchronous liver and lung metastases is estimated to be 11.4 months [2].

Due to the complexity of these patients, especially if synchronously diagnosed with the primary tumor in situ, decisions about selection and timing for surgical resection should be managed in the setting of a multidisciplinary team (MDT). Previous studies have shown that a low proportion of patients, about one-third, complete the initially intended curative resections due to disease progression [7,10]. These studies are based on the already selected patients referred to a liver MDT; thus, the actual proportion of patients presenting with synchronous liver and lung metastases and the resection rates are still unknown.

The aim of this nationwide registry-based study was to report on the incidence of synchronous liver and lung metastatic CRC, the proportion of patients undergoing metastasectomy and survival associated with different treatment approaches.

## 2. Materials and Methods

The present study is a population-based cohort study that includes all patients diagnosed with CRC in Sweden between the years 2008 (9.2 million inhabitants) and 2016 (9.9 million inhabitants), utilizing data from several nationwide registers in Sweden. The overall incidence of CRC in Sweden decreased in the past decade, but in patients under 50 years of age, the incidence of CRC continued to increase over time [11]. Colon resections are performed at 47 hospitals, while rectal cancer is resected at 31 hospitals. Referral to a liver or thoracic MDT meeting is decided at the local colorectal cancer MDT. In Sweden, liver and lung resections are regionally centralized to six University hospitals, each providing weekly held liver and lung-specific MDT meetings.

Swedish Colorectal Cancer Registry: All patients diagnosed with CRC and registered in the Swedish Colorectal Cancer Registry (SCRCR) between 2008 and 2016 while living in Sweden were extracted from the registry. The SCRCR is a nationwide quality registry that includes data on all patients diagnosed with CRC. The completeness, defined as the proportion of all cases registered in the SCRCR of all CRCs, is assessed annually by comparison to the Swedish Cancer Registry with an estimated overall completeness of 98.8% [12]. The registry contains data on date of diagnosis of the primary tumor, preoperative investigations and findings including the presence of liver and or lung metastasis, site and stage of the CRC, operative treatment of the primary tumor, histopathologic examination including distant metastasis (liver, lung and other locations) and data on the postoperative course after resection of the primary. The accuracy of the registration of synchronous metastases was recently validated and found to be high, where synchronous metastases were wrongly registered in 3.6% and not registered in 1% [13].

National Patient Register: The personal identity number assigned to all residents in Sweden enables linkage among different national registries. To identify the whole cohort of interest consisting of patients with simultaneous liver and lung metastases, data from SCRCR were linked to data from the National Patient Register (NPR). Data from all hospitalizations in Sweden are included in NPR, and reporting is obligatory in both public and private healthcare facilities. The information available in NPR includes all inpatients’ and outpatients’ visits including date of admission, date of discharge, main diagnosis, secondary diagnosis enabling up to 21 accompanying diagnoses during the intended hospitalization period, and data on procedures. From NPR, patients with liver metastases (C78.7) and lung metastases (C78.0) as the main or secondary diagnosis documented on an inpatient or outpatient visit within six months prior to and six months after the date of diagnosis of primary CRC were identified and included in the study cohort. The codes used for identifying other metastatic sites were as follows: pleura (C78.2), other respiratory organs (C78.1/.3), peritoneum (C78.6), other gastro-intestinal (C78.4/.5/.8), urinary system (C79.0/.1), skin (C79.2), nervous system (C79.3/.4), bone (C79.5), ovary (C79.6), adrenal (C79.7), and other specified (C79.8).

National Quality Registry for Liver, Bile Duct and Gallbladder Cancer: In the National Quality Registry for Liver, Bile Duct and Gallbladder Cancer (SweLiv), one of the sections registers the surgical interventions in the liver and includes data on the metastatic burden in the liver (number of metastases, size of largest metastasis, involved segments), detailed data on the intervention (resection and or ablation) and intervention-related complications. For patients with CRC metastatic disease, only patients undergoing intervention are included. Patients who underwent a liver intervention were thus found in SweLiv, and all available data concerning the liver metastatic burden and procedure-related data were extracted. The registry has a 97% nationwide coverage for inclusion [14].

National Quality Registry on Thoracic Surgery: In the same way, ThoR—a National Quality Registry on Thoracic Surgery—exists and contains data on surgical procedures in the lungs with a last reported nationwide coverage of 92.5% (primary and secondary tumors) [15]. This registry only contains information on patients undergoing a thoracic intervention, and if so, data were extracted.

To account for interventions not registered in SweLiv or ThoR, procedural codes linked to liver or thorax were extracted from NPR and merged into the dataset. Lung interventions (resection and/or ablation) and liver interventions (resection and/or ablation) were identified by the appropriate codes, and both liver and lung intervention codes were presumed to correspond to metastasectomy, but information on intent (curative or not) and if complete metastasectomy was performed cannot be interpreted from NPR.

Metastases detected within 6 months prior to and 6 months after the documented diagnosis date of primary colon or rectal cancer were considered synchronous. Liver and lung metastases were labeled as simultaneously diagnosed if both metastatic sites were diagnosed within the above-mentioned timeframe. Primary tumors of the caecum to the transverse colon were assigned as right-sided colon tumors, whilst tumors in the splenic flexure to sigmoid colon were assigned as left-sided colon tumors. Major hepatectomy was defined as resection of 3 or more liver segments according to Couinaud’s classification [16].

Descriptive statistics were used, and categorical variables are described using proportions and compared using the Chi-squared test or Fisher’s exact test. Continuous variables are described as median values (interquartile range (IQR) and min, max) if non-normally distributed and compared using Kruskal–Wallis equality-of-populations rank test (three-group comparison). Changes in treatment trend was evaluated using the Chi-square test for trend (denominator: all patients with liver and lung metastases). The main outcome was survival estimated from the diagnosis of liver metastases to date of death or last follow-up up set to 5 September 2019. The reverse Kaplan–Meier method was used to assess median follow-up [17]. Univariable survival estimates were illustrated in Kaplan–Meier graphs and compared using the log-rank test. Statistical significance was set to *p* < 0.050. STATA version 15.0 (StataCorp, Collage Station, TX, USA) was used for all data analyses. This study was approved by the Ethics Review Board in Linköping, Sweden. Because this study was a register-based study, individual informed consent was not required.

## 3. Results

### 3.1. Participants and Baseline Characteristics

In all, 60,734 individuals were diagnosed with colorectal cancer in Sweden between 2008 and 2016. Of these, 2703 (4.5%) were identified as diagnosed with liver and lung metastases within 6 months prior or after the diagnosis of colorectal cancer (Figure 1). Among these patients, 780 (29%) were also diagnosed with extrahepatic, non-pulmonary metastases. When excluding these, 1923 (3.2%) remained for further analysis, as illustrated in Figure 1. Patient and primary tumor characteristics are summarized in Table 1.

### 3.2. Treatment of Liver and Lung Metastases

Liver resections were conducted on 156 patients (based on data from NPR) of which 143 patients were identified in SweLiv and metastasis-specific data could be extracted for 35 patients undergoing complete metastasectomy and for 48 patients having liver resection and resection of the primary tumor only (Figure 1). The majority, 84 patients (54%), underwent liver resection on one occasion, while 46 patients (29%) had repeat hepatectomy and a further 27 patients (17%) underwent liver resection at three or more occasions. Lung resections were performed on 61 patients of which 44 patients were found in ThoR and metastasis-related data could be extracted on 32 patients having complete metastasectomy (Figure 1).

There were 44 (2.3%) patients having surgery for both liver and lung metastases and resection of the primary tumor (Figure 1). This subgroup, undergoing complete metastasectomy (*n* = 44), included patients who were younger and less often had a right-sided primary tumor compared to patients having liver resection only (*n* = 83) (Table 2). The subgroup where only the primary tumor was resected (no metastasectomy, *n* = 594) had a higher American Society of Anesthesiologists (ASA) score and a more advanced primary tumor stage as compared to the two other treatment groups depicted in Table 2. Only 58 patients (10%) among those who did not undergo any metastasectomy were referred for metastatic surgery (Table 2). Multiple liver metastases did not preclude patients from hepatectomy (Table 2). Four patients in the “liver resection and resection of primary only” group did not undergo liver resection per se but instead underwent thermal ablation. If these patients were too frail to undergo resection or if the MDT decided on ablation for any other reason cannot be interpreted from the register data (Table 2). The complexity of treatment strategy in metastatic CRC is illustrated by the multiple treatment allocations found in this cohort (Table 3, Figure 1).

### 3.3. Survival

The median follow-up from diagnosis of liver metastases was 10 months (range 0.03–142 months, IQR 135 months) and median follow-up of patients who were alive at end of follow-up was 53 months (range 29–142 months, IQR 103 months). Estimated 3- and 5-year OS following resection of liver and lung metastases (including resection of primary tumor) was 93.2% (95% CI 80.3–97.8%) and 74.2% (95% CI 57.2–85.3%), respectively (Figure 2). Patients undergoing liver resection and resection of their primary tumor had a significantly better survival compared to those only undergoing resection of the primary tumor, an estimated 5-year OS of 29.3% (95% CI 19.2–40.0%) versus 2.6% (95% CI 1.5–4.2%), *p* < 0.001, despite not having resection of the present lung metastases, as illustrated in Figure 2.

Twenty-six patients had liver resection only with an achieved estimated 3- and 5-year OS of 40% (95% CI 21–58%) and 5% (95% CI 0–21%), respectively (Table 3). The reason for not proceeding with complete metastasectomy and resection of the primary cannot be ascertained from the registries. Likewise, 13 patients underwent lung resection and resection of the primary CRC (Table 3) with 3- and 5-year OS of 85% (95% CI 51–96%) and 73% (95% CI 25–91%), respectively, in whom the extent of the liver metastatic burden, administration and response to chemotherapy remains unknown.

### 3.4. Trend over Time

The annual number of patients diagnosed with synchronous liver and lung metastases are depicted in Figure 3. Divided into three-year periods, no significant increase was seen in the proportion of patients undergoing complete metastasectomy when comparing the first time period (2008–2010) to the last time period (2014–2016) with 1.5% and 2.7%, respectively (*p* = 0.107) nor between the second (2011–2013) and last time period (2.6% versus 2.7%, *p* = 0.845). Referral for metastasectomy, by treating colorectal surgeon and/or medical oncologist, did however increase over time, from 7% (*n* = 44) in the first time period to 13% (*n* = 86) in the second time period and 22% (*n* = 145) in the last time period (*p* < 0.001). A clear trend over time was a decrease in the proportion of patients undergoing resection of the primary tumor only, in the presence of synchronous liver and lung metastases, from 46% in the first time period to 20% in the last time period, *p* < 0.001 (Figure 3).

The median survival of the entire group, irrespective of treatment, increased from 8 months (95% CI 6.7–9.1 months) in the first time period (2008–2010) to 10.5 months (95% CI 9.3–11.7 months) in the second time period (2011–2013) and 11.3 months (95% CI 10.3–13.1) in the third time period (2014–2016), with a significant increase in median survival comparing time period 1 and 3, *p* = 0.001 (Appendix A).

### 3.5. Regional Differences

The percentage of patients receiving complete metastasectomy ranged from 0.7% to 3.8% between the six healthcare regions of Sweden, as illustrated in Appendix A. There was a significant difference in resection rate between the regions with the highest and lowest resection rates, *p* = 0.007.

## 4. Discussion

This nationwide registry-based study demonstrates several intriguing findings. First, isolated synchronous liver and lung metastases are diagnosed in 3.2% of patients with CRC. Second, among them, only 2.3% undergo complete metastasectomy. When complete metastasectomy was performed, it resulted in excellent estimated long-term survival of 74% at 5 years. Third, an intermediate survival was seen in patients undergoing resection of liver metastases only, even when the lung metastases were not resected. Fourth, contrary to what was expected, this study did not show an increase in resection rate over time but revealed a low referral rate for metastasectomy and regional differences in resection rates within Sweden.

The proportion of synchronous liver and lung metastases aligns with previous findings of 3.1–3.4% [2,18]. The actual resection rate of both liver and lung metastases in a population-based setting has not previously been reported on. The low resection rate presented in this study of 2.3% is perceived as unexpectedly low. Most other studies on resection rates originates from surgical cohorts naturally affected by selection bias, also including both synchronously and metachronously detected liver and lung metastases and most often with an already resected primary tumor [7,8,9,10]. In these studies, about one-third of patients referred for the metastasectomy of simultaneously diagnosed liver and lung metastases underwent the intended curative treatment [7,10].

The reason for the low resection rate presented in this study and whether the decision on resectability was justified or not cannot be deduced from the registries. A limitation of the study is that despite the high degree of coverage in the Swedish registries, the registries do not provide detailed information on reasons to deny surgical treatment. Certainly, as a proof of selection, the group that underwent liver resection was younger with lower ASA, presented with a less advanced primary tumor stage and was more often located in the left colon and rectum. Nevertheless, a non-negligible proportion of patients who underwent liver surgery in this study had multiple liver metastases and subsequent major hepatectomy, which is in line with the long-known fact that resectability is not determined by the number and size of liver metastases but rather a sufficient future remnant liver volume [5]. The study draws attention to a low referral rate for metastasectomy; hence, one could hypothesize that not all patients eligible for metastasectomy are properly assessed for surgery. Generally, referral practice to regional MDT varies widely, from mandatory referral of all patients to referral at the discretion of referring physicians [19]. Medical oncologists and colorectal surgeons assess the resectability of liver metastases differently [20,21,22,23]. Reassessment of resectability by a hepatobiliary surgeon has shown that a meaningful number of patients with liver metastases are not managed according to the best available evidence, and the potential for higher resection rates is substantial [24,25]. Clearly, there is a need for an individualized, multidisciplinary approach to handle the complex decision-making process of patients with synchronously diagnosed liver and lung metastases, especially with the primary tumor in situ.

Even though no randomized trial has been performed on the topic, it is widely presumed that metastasectomy of both liver and lung metastases generates superior survival. Consistent with several other studies, a high estimated survival rate of 74% at 5 years was achieved among those selected to undergo complete metastasectomy in this study [7,9,23,26,27]. Contrary, the non-metastasectomy cohort, of which an unknown proportion had received palliative chemotherapy, had an estimated 5-year survival of 2.6%. The assumption that the surgical removal of lung metastases favorably affects survival has been questioned through the results from the randomized trial of Pulmonary Metastasectomy in Colorectal Cancer (PulMiCC) [28]. From that trial, it became clear that the assumption of zero survival without metastasectomy is contradicted and that the survival difference varies little, if any, in patients randomly assigned to metastasectomy compared to no surgical treatment of isolated lung metastases [28]. As the cohort of the PulMiCC trial only included patients with resectable lung metastases, with previously resected CRC, no concurrent liver metastases and by being considered for metastasectomy, hence presumably having favorable features, it is unclear if the results from the PulMiCC trial can be applicable on patients suffering from synchronously diagnosed liver and lung metastases. Instead, the results from the PulMiCC trial can support that the theory of lung metastases themselves may not present the decisive factor for survival and thereby supporting the suggestion presented by Mise et al. to resect liver metastases in selected patients with unresectable lung metastases yielding a survival benefit compared to palliative chemotherapy only [29]. This is further supported by the intermediate survival displayed in this population-based setting with a 5-year survival of 29% in patients having liver resection in the presence of synchronous lung metastases, even when the lung metastases were not resected, as opposed to 2.3% if not undergoing metastasectomy at all. On the other hand, an analysis based on the Surveillance, Epidemiology and End Results database, assessing the impact of metastasectomy in metastasized CRC patients with resected primary tumor, found a significant increase in survival for liver resection but not for metastasectomy of lung or both sites [30].

Because of improved surgical techniques and treatment possibilities, we expected an increased resection rate of both liver and lung metastases over time but that was not found in this study. A Dutch study evaluating nationwide trends in incidence and treatment between 1996 and 2011 found an increase in metastasectomy rate over the years but only in patients with metastatic disease confined to one organ, which was most evident in patients with isolated liver metastases [2]. The resection rate of multiple metastatic sites (not further specified) remained constant during the study period [2]. Whether treatment trends have changed during the last five years remains to be revealed.

The resection rate of the primary tumor in the presence of synchronous liver and lung metastases decreased over the study period, which was consistent with previous findings [2]. This decrease could be the result of recent publications addressing the question of whether to perform palliative tumor resection in incurable stage IV disease or instead favoring colonic stents and diverting stoma [31,32].

Over the last decades, thermal ablation has been established as a treatment alternative to liver resection, mainly for small liver metastases < 3 cm, as adjunct to liver resection for patients with multiple bilobar disease or as completion treatment [33,34,35]. As a fact, current treatment guidelines include thermal ablation as a treatment alternative to liver resection for selected oligometastatic colorectal cancer disease [36,37]. The use of thermal ablation in this cohort was limited and only registered in four patients as the sole treatment strategy. Perhaps, an extended utilization of thermal ablation could have increased the proportion of patients undergoing complete metastasectomy, especially in the frail subgroup of patients.

This study shows variation in the rates of complete metastasectomy across Sweden. Similarly, such variations have previously been shown for both lung metastasectomy and liver metastasectomy [38,39]. Despite statistical significance, these differences could be explained by the low number of patients having complete metastasectomy, but it requires reflection as Sweden, even though geographically large, has relative few inhabitants, and all six health units follow the same national guidelines for metastatic CRC [40].

The present study is hampered by several limitations. First, it is limited by its retrospective nature dating back to treatment prior to 2016; on the other hand, this allows for a relatively long follow-up, making the survival analysis reliable. Second, although the study managed to present population-based data including resection rate of both liver and lung metastases for the first time, the analyses are limited by the lack of completeness regarding patient and tumor-specific data from the registries, regarding the large group of non-resected patients. In addition, no reliable data could be obtained on stereotactic body radiotherapy as treatment of lung metastases, nor the proportion of the non-resected population having chemotherapy. Eligibility for liver and lung metastasectomy includes confirming operative candidacy, which is also unknown from this dataset, as is whether the patient was assessed by a dedicated liver and or lung multidisciplinary team which makes the reasons for the low resection rate and whether the presumed low referral was reasonable or not impossible to analyze. The low number of patients having complete metastasectomy and the even lower number of patients with metastasis-related data makes any attempt on further analysis in a multivariable regression model meaningless. These limitations can only be overcome by the review of medical records on all patients, which hopefully is a future study. Nonetheless, the results from this study are still relevant, as they demonstrate excellent survival for patients completely treated for synchronous liver and lung metastasis as well as colorectal primary.

## 5. Conclusions

In summary, we provide reliable population-based numbers on the incidence and curative treatment of synchronously diagnosed liver and lung metastatic CRC. We conclude that it is likely that a larger proportion of this patient cohort could be offered treatment that leads to a prolonged overall survival. For this reason, a larger proportion of this patient cohort should be referred and evaluated at a dedicated multidisciplinary conference with appropriate specialties attending.

## Figures and Tables

**Figure 1 cancers-15-01434-f001:**
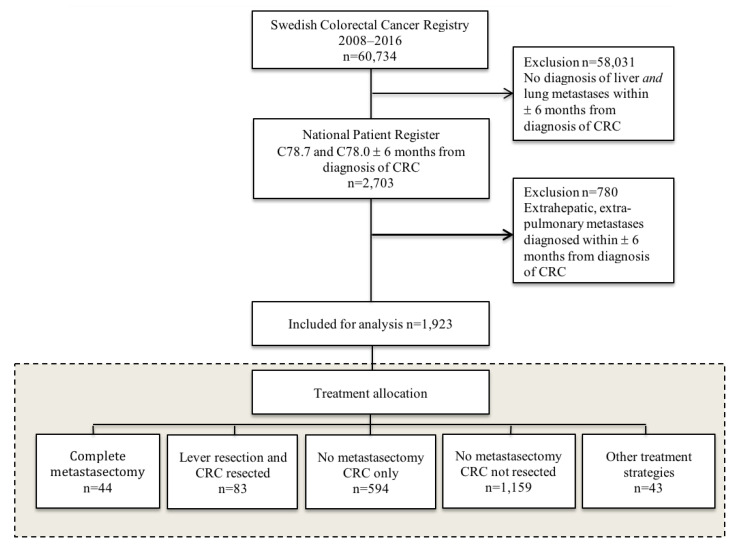
Flowchart of the identification of all patients diagnosed with liver and lung metastases within six months from diagnosis of colorectal cancer between 2008–2016 in Sweden. CRC, colorectal cancer.

**Figure 2 cancers-15-01434-f002:**
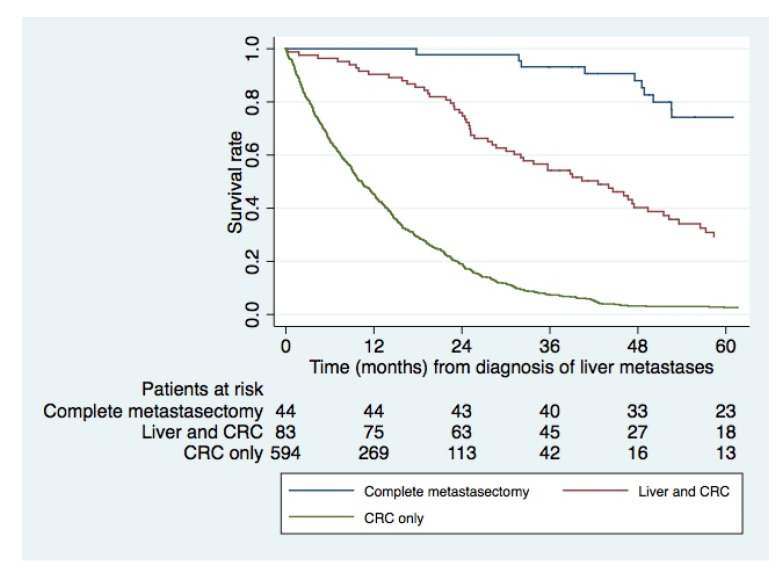
Kaplan–Meier estimates of overall survival in patients treated with complete metastasectomy versus liver resection only versus resection of primary only. Complete metastasectomy including resection of the primary tumor resulted in a 5-year overall survival (OS) of 74.2% (95% CI 57.2–85.3%), while patients having resection of liver metastases and primary tumor had a corresponding estimated median and 5-year OS of 43 months (95% CI 31–49 months) and 29.3% (95% CI 19.2–40.0%), respectively. Resection of the primary tumor only resulted in a median survival of 10 months (95% CI 9–12 months) and a 5-year OS of 2.6% (95% CI 1.5–4.2%). There was a significant survival difference between complete metastasectomy and resection of liver metastases only, log rank test *p* < 0.001 and between the latter and resection of the primary only (no metastasectomy), log-rank test *p* < 0.001. CRC, colorectal cancer.

**Figure 3 cancers-15-01434-f003:**
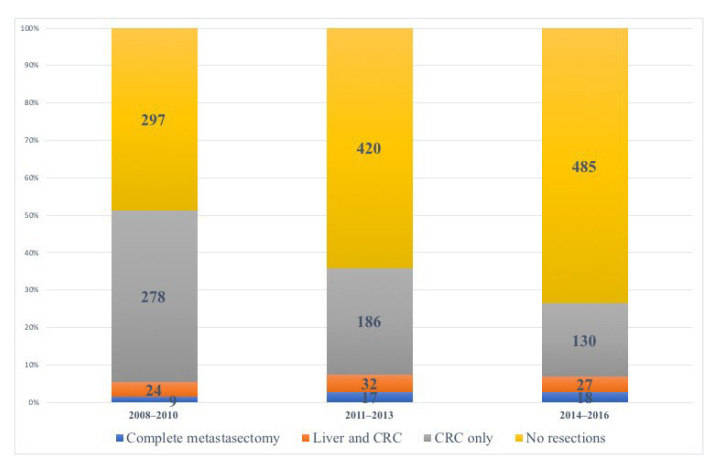
Treatment trends over time. Bar chart illustrating different treatment strategies in different time periods; 2008–2010 (*n* = 608), 2011–2013 (*n* = 655), and 2014–2016 (*n* = 660). Complete metastasectomy was conducted in 9 patients (1.5%) in the first time period, 17 patients (2.6%) in the middle time period and 18 patients (2.7%) in the last time period of the study. The proportion of patients having resection of the primary tumor only, in the presence of synchronous liver and lung metastases, significantly decreased over time from 46% (2008–2010) to 28% (2011–2013) and finally 20% (2014–2016), log-rank test *p* < 0.001. CRC, colorectal.

**Table 1 cancers-15-01434-t001:** Patient and tumor characteristics in 1923 patients with synchronously diagnosed liver and lung metastases from colorectal cancer.

	*n* = 1923 (%)
Patient characteristics	
Gender, *n* = 1818	
Male/Female	1024 (56)/794 (44)
Age (years), median (IQR)	70 (14)
ASA, *n* = 671	
1	68 (10)
2	323 (48)
3	232 (35)
4	48 (7)
Primary tumor characteristics	
Primary tumor location, *n* = 1912	
Caecum	206 (11)
Ascending colon	151 (8)
Hepatic flexure	73 (4)
Transverse colon	69 (4)
Splenic flexure	31 (2)
Descending colon	59 (3)
Sigmoid colon	515 (27)
Rectum	808 (42)
Resection of primary tumor	734 (38)
Pathological tumor stage, *n* = 527	
pT0	5 (1)
pT1	4 (1)
pT2	15 (3)
pT3	290 (55)
pT4	213 (40)
Pathological nodal stage, *n* = 506	
pN0	95 (19)
pN1	166 (33)
pN2	245 (48)
Referred for metastasectomy ^1^	275 (14)

^1^ Referred for metastatic surgery evaluation by treating colorectal surgeon and/or medical oncologist, data from the Swedish colorectal cancer registry. Values are *n* (%) unless otherwise indicated. IQR, interquartile range; ASA, American Society of Anesthesiology.

**Table 2 cancers-15-01434-t002:** Comparison of patient and tumor characteristics in three different treatment strategies.

	Liver and Lung Metastasectomy and Resection of Primary, *n* = 44	Liver Resection and Resection of Primary Only, *n* = 83	Resection of Primary Only, *n* = 594	*p* *,†
Age (IQR)	62 (13)	68 (13)	71 (55)	<0.001 ‡
Gender, female	21 (48)	33 (40)	272 (47)	0.477
ASA ^1^				
1	11 (25)	17 (21)	39 (7)	<0.001
2	22 (50)	39 (49)	255 (48)	
3	11 (25)	24 (30)	193 (36)	
4	0 (0)	0 (0)	47 (9)	
Missing	0	3	60	
Primary tumor location ^2^				
Right-sided colon	4 (9)	27 (33)	186 (31)	0.032
Left-sided	21 (49)	28 (34)	190 (32)	
Rectum	18 (42)	28 (34)	217 (37)	
Missing	1	0	1	
Tumor stage of primary ^2^				
T1–T2	4 (9)	5 (7)	8 (2)	<0.001
T3	30 (70)	51 (66)	201 (52)	
T4	9 (21)	21 (27)	181 (46)	
Missing	1	6	204	
Referral for metastatic surgery ^3^	37 (84)	53 (64)	58 (10)	<0.001
Number of liver metastases ^1^			N/A	
1	10 (29)	17 (36)		0.646 **
2–5	18 (51)	24 (50)		
6–10	5 (14)	3 (6)		
≥11	2 (6)	4 (8)		
Missing	9	35		
Liver resection ^1^			N/A	
Major hepatectomy	13 (32)	28 (37)		0.303 **
Minor hepatectomy	28 (68)	44 (58)		
Ablation only	0	4 (5)		
Missing	3	7		
Size of largest liver metastasis, mm (IQR) ^1^	20 (18)	25 (21)	N/A	0.016 ‡
Missing	9	12		
Number of lung metastases (min, max) ^4^	1 (1, 9)	N/A	N/A	
Missing	12			
Unilateral lung metastases ^4^	32 (100)	N/A	N/A	
Missing	12			

* *p* values refers to a comparison between all three groups, except ** which indicate a comparison between “complete metastasectomy” and “liver resection and resection of primary only”. ^1^ Non-complete data on patient and metastasis characteristics due to missing data in National Quality Registry for Liver, Bile Duct and Gallbladder Cancer (SweLiv). ^2^ Non-complete data on primary tumor location from Swedish Colorectal Cancer Registry. ^3^ Referred for metastatic surgery evaluation by treating colorectal surgeon and/or medical oncologist, data from the Swedish Colorectal Cancer Registry. ^4^ Based on metastasis data on 32 patients from National Quality Registry on Thoracic Surgery. Values are *n* (%) unless otherwise indicated. † Categorical variables were compared using the chi-squared test or Fisher’s exact test as appropriate. ‡ Continuous variables were compared using Kruskal–Wallis equality-of-populations rank test (three-group comparison) or Wilcoxon rank sum test (two-group comparison. IQR, interquartile range; ASA, American Society of Anesthesiology; N/A, not applicable.

**Table 3 cancers-15-01434-t003:** Treatment allocation in 1923 patients diagnosed with liver and lung metastases within six months from diagnosis of colorectal cancer in Sweden between 2008 and 2016.

Treatment Allocation	*n* = 1923
Liver + Lung + CRC	44 (2.3)
Liver + CRC	83 (4.3)
CRC only	594 (30.9)
No metastasectomy, no resection of CRC	1159 (60.3)
Liver + Lung	3 (0.2)
Liver only	26 (1.4)
Lung only	1 (0.05)
Lung + CRC	13 (0.7)

Values are *n* (%). CRC, colorectal cancer.

## Data Availability

Data are available through the different Swedish national quality registries and the National Patient Registry following appropriate approval. Derived data supporting the findings of this study are available from the corresponding author J.E. on request after appropriate ethical approval.

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
