# Peer review of "The Resection Rate of Synchronously Detected Liver and Lung Metastasis from Colorectal Cancer Is Low—A National Registry-Based Study"

_cancers, 2023, doi:10.3390/cancers15051434_

Round 1
Reviewer 1 Report
This paper researched the resection rate and survival rate of synchronously detected liver and lung metastasis. The theme is very important, but the methods and items to be considered should be revisited.
Major comments
#1. The method of patient selection is very important in this paper. The authors described the method of patient selection in the text, but it is confusing and should be illustrated in the study flow diagram.
#2. In P.4, the authors described that the subgroup of complete metastasectomy were younger, had a lower American Society of Anaesthesiologists (ASA) score, less often a right-sided primary tumour and had a less advanced primary tumour stage compared to patients having liver resection only (n=83) and no metastasectomy (n=594). I think this interpretation is incorrect about ASA, less advanced primary tumour stage. I think the subgroup of resection of primary only had a higher ASA score and advanced tumor stage compared with the other groups. The other two subgroups had no significant differences in this point.
Minor comments
#1. The descriptions were not organized in Table 2. In addition, the percentage of descriptions of "with simultaneous cauterization" is different from other items. It would be better to unify the population with other items.
#2. In Table 3, the number of populations is different for each item, but there were items that are not mentioned with respect to the number of populations. Please add the numbers.
#3. Some of the citations were in the wrong order. Please check.
Reviewer 2 Report
Dear Editor and Authors,
I read and evaluated this manuscript titled “The resection rate of synchronously detected liver and lung metastasis from colorectal cancer is low – a national registry-based study” by Dr. Engstrand and colleagues from Sweden.
In this nationwide, retrospective, database case series analysis the authors are aiming to present survival outcomes in patients with colorectal cancer and concurrent liver and lung metastasis. They are able to demonstrate that complete metastasectomy conferred better OS survival.
This is an interesting study centered on the fact that concurrent liver and lung metastasis from colorectal cancer is relatively rare. What is also rarer is the surgical management/metastasectomy of both sites. This is (as the authors themselves concede) not often done and certainly not established practice. To that extent I feel the manuscript has value to be presented to the scientific community – it is more evidence towards a more aggressive surgical management of these patients and I feel colorectal cancer patients in general for it provides much improved survival! This axiom maybe something we as oncological surgeons are well aware of but unfortunately evidence are lacking to back up our intuition, until possibly now I believe.
However, there is a significant limitation, inherent due to the nature of the pathology and quite impossible to address, that is the small sample number of patients analyzed!! At 44 patients the sample is small at best, especially if we consider it represents the whole of Sweden!!
This is overall a well conducted study, the methodology is clear and the analysis is as expected for such a type of work. The manuscript is well written in clear language with only minor English corrections required. It is well illustrated with tables and figures which are clear and understandable.
I only have a couple of minor points to make to the authors:
1. The number of patients having a complete metastasectomy is small at 44 patients and this is something that needs to be mentioned in the limitations section. It is understandable that this number cannot be increased!!
2. Why was a multivariable analysis not performed to elucidate factors/variables related to long term survival? Metastasectomy could be utilized as an individual study variable or in combination with other variables within the model!!
3. I noticed in table 3, in the liver and primary site only resection group, 4 patients (21%) had an RF ablation and not surgery per se. The authors need to make a small mention of this fact in their text along the context that a) either these patients were too frail to undergo surgery and b) the results in the literature when we are talking of a certain size tumor are comparable between the two techniques.
In conclusion, this is an interesting study as mentioned above and I feel with some minor editing would improve even more. It is not the strongest type of evidence out there, but compared to what is available on the matter its pretty good. Therefore, I am glad to recommend its publication following some minor corrections. Kind regards to all and good luck.
Round 2
Reviewer 1 Report
After the correction, the paper is much better, but needs a few more revisions.
Major comments;
#1. I think the introduction of the flowchart is very good, but there are a few things that need to be corrected. First, I think the full spelling of NPR should be shown in the figure legend. Second, the wording at the bottom of the figure is confusing and should be corrected. A total number of the patients of complete metastasectomy was 44, although 35 patients undergoing complete metastasectomy were identified in SweLiv and 32 patients having complete metastasectomy were identified in ThoR. I think there were probably several duplicate cases, but I didn't know how to exclude them. One option might be for the authors to delete the last line, because It's a misleading representation.
#2. I think it is a good idea to correct the interpretation in Table 2.
Minor comments;
#1-2. I understand that in Table 2, the data on patient and metastasis characteristics were missing and were not complete. I only recommended that the number of each item be shown. The number of studies was written only for the item on the numbers of liver metastases, but not for the other items. Conversely, I found it curious why only one item showed a total number.
